# Long-Term Survival after Acute Myocardial Infarction in Lithuania during Transitional Period (1996–2015): Data from Population-Based Kaunas Ischemic Heart Disease Register

**DOI:** 10.3390/medicina55070357

**Published:** 2019-07-09

**Authors:** Ricardas Radisauskas, Jolita Kirvaitiene, Gailutė Bernotiene, Dalia Virviciutė, Ruta Ustinaviciene, Abdonas Tamosiunas

**Affiliations:** 1Department of Environmental and Occupational Medicine, Medical Academy, Lithuanian University of Health Sciences, LT-47181 Kaunas, Lithuania; 2Institute of Cardiology, Medical Academy, Lithuanian University of Health Sciences, LT-50103 Kaunas, Lithuania; 3Department of Preventive Medicine, Medical Academy, Lithuanian University of Health Sciences, LT-47181 Kaunas, Lithuania

**Keywords:** acute myocardial infarction, long-term survival, sex, age, register

## Abstract

*Background and Objective*: There is a lack of reliable epidemiological data on the long-term survival after acute myocardial infarction (AMI) in the Lithuanian population. The aim of the study was to evaluate the long-term (36 months) survival after AMI among persons aged 25–64 years, who had experienced AMI in four time-periods 1996, 2003–2004, 2008, and 2012. *Material and Methods*: The source of the data was Kaunas population-based Ischemic heart disease (IHD) register. Long-term survival after AMI (36 months) was evaluated using the Kaplan–Meier method. The survival curves significantly differed when *p* < 0.05. Hazard ratio for all-cause mortality and their 95% CIs, adjusted for baseline characteristics, were estimated with the Cox proportional hazards regression model. *Results:* The analysis of data on 36 months long-term survival among Kaunas population by sex and age groups showed that the survival rates among men and women were 83.4% and 87.6%, respectively (*p* < 0.05) and among 25–54 years-old and 55–64 years-old persons, 89.2% and 81.7%, respectively (*p* < 0.05). The rates of long-term survival of post-AMI Kaunas population were better in past periods than in first period. According to the data of the Kaplan-Meier survival analysis, long-term survival of 25 to 64-year-old post-AMI Kaunas population was without significantly difference in 1996, 2003–2004, 2008 and 2012 (Log-rank = 6.736, *p* = 0.081). The adjusted risk of all-cause mortality during 36 months among men and 25 to 54-year-old patients was on the average by 35% and 60% lower in 2012 than in 1996, respectively. *Conclusion*: It was found that 36 months survival post MI among women and younger (25–54 years) persons was significant better compared to men and older (55–64 years) persons. Long-term survival among 55 to 64-year-old post-AMI Kaunas population had a tendency to decrease during last period, while among 25–54 years old persons long-term survival was without significant changes. The results highlight the fact that AMI survivors, especially in youngest age, remain a high-risk group and reinforce the importance of primary and secondary prevention for the improvement of long-term prognosis of AMI patients.

## 1. Introduction

Ischemic heart disease is currently at the forefront cause of death in Europe [1]. Although the case-fatality rates of acute myocardial infarction have been declining during the last thirty years, death rates from IHD in Lithuania is one of the biggest in Europe [2,3]. Some studies from various European countries showed declining trends in cardiovascular mortality due to changes in cardiovascular risk factors and to improved treatment [4,5].

The analysis of the data of 39 studies on five-year mortality rates after AMI showed that during the period of 1966–2012, long-term mortality after AMI was 58.6% for women and 45.6% for men in Scotland, 45.2% for women and 33.7% for men in the Netherlands, 35.0% for women and 23.0% for men in Spain, and 35.7% for women and 36.3% for men in Sweden [6]. In Australia, monitoring of long-term survival after AMI during the period of 2003–2009 demonstrated that the risk of re-myocardial infarction was higher and long-term survival was lower among women compared to men in all age groups [7].

Very relevant factors of AMI care are time for early diagnosis, the fastest reperfusion time, and the use of different types of medications (antiplatelet agents, antihypertensives, antianginal drugs, and lipid-lowering drugs) [8,9,10,11,12]. Previous surveys from some European countries estimated an enrichment in coronary care, accentuating improve of invasive diagnostics and treatment procedures, and an extensive use of varies medications [6,13,14,15,16,17]. Meanwhile, there was no observed marked improvement in short- or long-term survival among persons who had experienced AMI in Lithuania or other Baltic countries [13,18]. The latest data of coronary care quality and mortality trends due to AMI in Central and Eastern European countries showed lack of similar data on extensive changes of acute coronary care in these countries [17,19]. According to official Lithuanian mortality statistics, every eighth death from ischemic heart disease occurs in persons of working (25–64 years) age, of whom even a quarter in working age men [1].

The aim of the study was to evaluate the long-term (36 months) survival after AMI among persons aged 25–64 years who had experienced AMI in four time-periods 1996, 2003–2004, 2008, and 2012.

## 2. Materials and Methods

### 2.1. Study Sample

The source of data was Kaunas population-based Ischemic heart disease register, which take part among 25–64 years-old Kaunas city (Lithuania) inhabitants. The source of the population data was the Lithuanian Statistical Department, which yearly publishes reports on the population size. The size of the Kaunas population aged 25–64 years was 221, 404 persons in 1996, 399, 442 in 2003–2004, 189, 220 in 2008 years, and 165, 156 in 2012.

The methods used for the data collection were those applied by the WHO for the international MONICA (Monitoring of Trends and Determinants in Cardiovascular Diseases) project [20] and were described in detail elsewhere [21]. All permanent residents of Kaunas aged 25–64 years who had experienced AMI in 1996, 2003–2004, 2008, and 2012 have been considered in the present study. The diagnosis of AMI and quality control procedures was based on the criteria proposed by the WHO MONICA project [20]. The “cold pursuit” technique (i.e., retrospective data collection) was used to identify and verify AMI events [20]. Multiple sources of information (hospital discharge records, hospital health records, records of outpatient departments, autopsy, and medico-legal records) were used for case ascertainment, including death certificates. All suspected AMI events were recorded on special forms translated from the Acute Myocardial Infarction Events Registration Form of the WHO MONICA project. The epidemiological diagnostic category of AMI or coronary death was defined referring to four diagnostic criteria: (1) symptoms of a coronary event, (2) dynamic changes of electrocardiogram (ECG) indicating the development of AMI, (3) changes of cardiospecific enzymes activity in blood serum, and (4) necropsy findings [20]. For verification of AMI cases four epidemiological diagnostic categories (EDC) were set and coded as “definite AMI”, “possible AMI” (or, in case of death, “possible coronary death”), “not AMI” (or the cause of death was not coronary artery damage), and “insufficient data to define a category”. Cases with the following epidemiological diagnostic categories were included into the list of the study subjects: “definite AMI” and “possible AMI” (for survivors), and “definite AMI”, “possible coronary death”, and “insufficient data to define a category” (for patients who died). Every AMI event had to have its apparent onset within the study period and had to occur more than 28 days after any previously recorded AMI event in the same subject. Multiple AMI attacks occurring within 28 days from the onset of the symptoms of the first attack were considered as one event. An AMI event was defined as fatal if death occurred within the first 28 days from the onset. If the patient was alive after 28 days from the onset of the attack, the AMI was classified as non-fatal. All patients residing permanently in Kaunas city and suspected of having died from AMI or having had a non-fatal AMI were registered.

Throughout the study period, the same methods for the identification of AMI cases, the same diagnostic criteria of AMI, and the same quality assurance procedures were applied in order to assure comparability of the data. All data were evaluated by sociodemographic variables as sex and age. The study also assessed data by the final clinical diagnosis of events using codes of International Classification of Diseases (ICD) (AMI: ICD-9 codes 410 and ICD-10 codes I21–I22 and unstable angina pectoris (AP): ICD-9 codes 411 and ICD-10 codes I20.0) and by EDC classification (“definite AMI” and “possible AMI”). Patients who had experienced AMI were categorized as ST elevation myocardial infarction (STEMI) or non-ST elevation myocardial infarction (NSTEMI) based on their ECG changes.

In our study comorbidities as arterial hypertension (AH), stroke, diabetes mellitus (DM), previous MI, previous acute heart failure, smoking, and overweight were also registered and hospital bed-days was evaluated from the patient hospital health records.

In 1996, 2003–2004, 2008, and 2012, a total of 303, 797, 383, and 335 cases of AMI respectively were registered. In our study some degree of misclassification is apparently unavoidable, but is likely to be minor and not influence our findings.

The study protocol was approved by the Lithuanian Bioethics Committee (ref. No. 14-27/03 December 2001). All patient records/information were anonymized and de-identified prior to the analysis.

### 2.2. Ascertainment of Outcome Events

Information on the causes of death of 25 to 64-year-old inhabitants of Kaunas city who died during 1996–1999, during 2003–2007, during 2008–2011, and during 2012–2015 was obtained from Kaunas Civil Registry Office. All medical death certificates were reviewed to verify the diagnosis. During 1996–1999, causes of death were coded by using the International Statistical Classification of Diseases and Related Health Problems, the 9th revision (ICD-9), and during 2003–2007, 2008–2011 and 2012–2015, causes of death were coded by using the International Statistical Classification of Diseases and Related Health Problems, 10th revision (ICD-10). Certificates with the diagnosis of IHD were selected for verification (ICD-9 codes 410–414, and ICD-10 codes I20–I25). Following the WHO recommendations, in order to ascertain all cases of death, medical death certificates that stated the following causes of death were selected: diabetes mellitus (ICD-9 codes 250, and ICD-10 codes E10–E14), obesity (ICD-9 codes 278, and ICD-10 codes E65–E68), dyslipidemia (ICD-9 codes 272, and ICD-10 codes E78), hypertension (ICD-9 codes 401–405, and ICD-10 codes I10–I15), other heart diseases (ICD-9 codes 420–429, and ICD-10 codes I30–I52), cerebral vascular damages (ICD-9 codes 430–438, and ICD-10 codes I60–I69), diseases of arteries, arterioles, and capillaries (ICD-9 codes 440–449, and ICD-10 codes I70–I77), and unclear causes of death (ICD-9 codes 797–799, and ICD-10 codes R95–R99).

During the period of 1996–1999, 60 persons died in the group of 25 to 64-year-old persons who had experienced AMI in 1996. During the periods of 2003–2007 and 2008–2011, 118 and 50 persons, respectively, died in groups of persons who had experienced AMI during 2003–2004 and in 2008. During the last analysed period (2012–2015), 53 persons died in the group of 25–64 years who had occur AMI in 2012. Only the first registered cases of AMI were analyzed. The life status of persons with AMI and surviving 28 days was followed up and checked using the National database of the register of deaths and their causes. Every person who suffered from AMI was followed-up to death, but no longer than 36 months. Those who survived 36 months or more, or who died later than 36 months after the onset of the AMI, are included in the analysis as alive (censored). Patients with no record of death were censored at 31 December 2015. In this survey, all-cause mortality within 36 months after AMI was studied. In this study, long-term survival (36 months) after AMI was analysed and assessed among persons from the onset of AMI during four study time-years periods.

### 2.3. Statistical Analysis

Categorical variables were summarized by proportions expressed in percentages and compared by the chi-square test. Continuous variables were summarized by means, standard deviations, and confidence intervals (CI). Changes in time (between time-years periods) were calculated the level of significance for trend (for continuous variable—using ANOVA and for categorical variables using the method of linear regression on logarithms of the percentage). Differences of 2 proportions were assessed by z test calculation and comparisons of multiple percentages were used Bonferroni calculation. Long-term survival after AMI was analyzed using the Kaplan–Meier method. The Log-rank criterion was applied for comparing the survival curves. A survival curves significantly differed if the Log-rank criterion *p* was < 0.05.

Cox proportional hazards regression analyses to identify the hazard of all-cause mortality at 36 months follow-up in patients who had experienced AMI in four time-periods (1996, 2003–2004, 2008, and 2012) by sex and age were used. For the logistic regression model, we considered that 1996 year was as reference. In addition to crude mortality rates, the subjects were adjusted by baseline characteristics (sex, age, clinical diagnosis, EDC classification, STEMI, AH, overweight, diabetes, previous acute heart failure, previous MI, stroke, smoking). The strength of the association between the studied years and mortality rates was examined using the single variable age-adjusted and multivariable-adjusted incidence hazard ratios (HR) and corresponding 95% CI. A two-sided *p* value < 0.05 was considered statistically significant. The statistical analysis of the data was performed using the software packages SPSS 20.0 and MS Office.

## 3. Results

Baseline characteristics of the study are presented in Table 1. The proportion of men and older (55–64 years-old) persons in the study sample was without significantly changes from 1996 to 2012, and there were no significant changes in the frequency of most comorbidities. Assessing data by patient’s clinical diagnosis during analysed time-period cohorts were estimated significantly higher events with diagnosis of AMI and significantly less events with diagnosis of unstable AP in 1996 cohort than in 2008 cohort as no significant differences in other time-periods cohorts were found. Evaluating data by the EDC classification were found to significantly increase the proportion of “definite AMI” cases and significantly less “possible AMI” events only in 2012 cohort compared to the 1996 cohort. The results showed no significant changes in the proportion of patients with STEMI during the study period, while the proportion of patients with AH during the study period significantly increased from 65.7% to 83.0% (*p* = 0.02) and the frequency of previous MI significantly decreased from 27.1% to 22.4% (*p* = 0.03). From 1996 to 2012, the mean duration of hospital stay significantly decreased too. Although the trend of men during the four study periods was without significant changes but was found that the proportion of men (77.2%) in 1996 was significantly higher than in 2008 (69.2%) (*p* < 0.05). The frequency of diabetes in 1996 was significantly lower than in the years 2003–2004 and 2012, while the incidence of previous acute heart failure in patients with MI was significantly higher in 1996 compared with 2008. Therefore, the trend of smoking rates during investigated period was without significant changes smoking rate in 1996 were the highest and amounted to 39.6% and significantly differed from the result of 2012 (30.1%) (*p* < 0.05). Significantly lowest rate of overweight among persons with AMI was found in the period 2003–2004, and the highest overweight rate in 2008, respectively 60.6% and 67.6% (*p* < 0.05).

Baseline characteristics of patients with AMI by age are presented in Table 2.

Data on age groups showed that in both the younger (25–54 years-old) and older age (55–64 years) cohorts the proportion of cases with clinical diagnosis of AMI in 1996 cohort, there was a significant more, while events with unstable AP was significant less compared to the other time-period cohorts.

Evaluating data by the EDC classification in age groups were found a significantly less proportion of events for “definite AMI” cases in the younger age (25–54 years-old) group in 1996 comparing with 2008 and 2012 cohorts, therefore in oldest age (55–64 years) group significantly changes not found. In “possible AMI” age groups some significant changes were estimated. In 1996 the percentages of “possible AMI” events were significantly higher compared with 2008 and 2012 cohorts in both age groups. During the study period, the proportion of cases of STEMI was without significantly changes among both patients aged 25–54 and 55–64 years, meanwhile older persons who experienced MI in 1996 was significantly more than in 2008 years, respectively 40.9% and 30.0% (*p* < 0.05). The proportion of other comorbidities and such risk factors of cardiovascular disease (CVD) as AH and diabetes significantly increased among 55 to 64-year-old and 25 to 54-year-old patients respectively during 1996–2012. The proportion of previous MI and previous acute heart failure tended to decrease only among younger (25–54 years) persons, than among older persons was without significant changes as the rate of diabetes among those during study years. During the study period, the proportion of smoking among patients aged 25–54 years tended to decrease, while it was without significantly changes among 55 to 64-year-old patients. The frequency of overweight among both 25–54 and 55 to 64-year-old persons who experienced MI was without significant changes too. The means of hospital bed-days among both age groups significantly decreased from 1996 to 2012. Despite of the fact that the stroke rate did not change significantly during the study period, the lowest percentage (0.7%) was determined in 2008 and significantly differed from the period of stroke in the period 2003–2004 where its frequency was (2.6%). Comparing separate study periods, it was found that significantly higher rate (12.4%) of previous acute heart failure were detected in 1996 compared to 2003–2004 and 2008, respectively 7.1% and 7.3%. Although the overweight trend was without significant changes among younger and older investigated persons who had have MI but the significant highest percentage of overweight among younger subjects was found in the last study (2012) period (69.2%), while among elderly subjects in the first study (1996) period (68.8%).

Data on 25 to 64-year-old Kaunas population who had experienced AMI in 1996, 2003–2004, 2008, and 2012, and survived or did not survive for 36 months by sex, age, and survival time are presented in Table 3. During the analyzed period (1996–2012), 84.5% of persons (83.4% of men and 87.6% of women, *p* = 0.03) survived for 36 months after AMI.

Comparing 36 months survival after MI in 1996 and 2008 significant differences were found (Log-rank = 5.894, *p* = 0.015). In the 2008 cohort, the survival of 36 months after MI was significantly better than in the cohort of 1996. There were no significant differences in the sexes, while significant differences in survival of 36 months after AMI were observed in the age groups. In the older age group (55–64 years), the 36-month survival rate for AMI was significantly better in 2008 cohort compared to the cohort of 1996 (Log-rank = 6.664, *p* = 0.01), while no significant differences were found in the younger (25–54 years) age group.

Comparing the separately cohort of the subjects, significant differences in 36 months survival after AMI were found. In the cohort of persons with AMI in 2003–2004 and in 2008, the 36 months survival after AMI was significantly better than that in 1996 cohort (80.2% and 85.2% (*p* = 0.04), and 86.9% (*p* = 0.02) respectively. Meanwhile, between the 36-month survival after AMI in 2012 cohort compared to the 1996 cohort, no significant differences were found.

There were no significant differences between men and women comparing 36 months survival after AMI in the separated 4 time-periods. According to the data of the IHD register, from 303 inhabitants of Kaunas city aged 25–64 years who had experienced AMI in 1996 survived for 36 months 79.9% men and 81.2% women (*p* > 0.05). It is interesting to note that during the remaining three study periods there was no significant difference in the 36 months survival after MI rates between men and women (Table 3). In both men and women, the 36 months survival after AMI compared 1996 years cohort to with subsequent separately cohorts (2003–2004, 2008, and 2012 years) no significant differences were found.

The analysis of data on long-term survival among Kaunas population by age groups (25–54 and 55–64 years) showed that in age groups of 25–54 and 55–64 years, survival rates was 89.2% and 81.7%, respectively (*p* = 0.0001).

There were observed significant differences only in oldest (55–64) age group comparing 36 months survival after AMI in the separated 4 time-periods. The survival of 55 to 64-year-olds for 36 months survival after AMI was significant better to those who experienced AMI in 2008 compared with the 1996 cohort, respectively 85.8% and 76.3% (*p* = 0.01), therefore there were no significant differences in other cohorts. Significant differences were found between younger and older subjects in separated cohorts of the study year when evaluating and comparing data for 36 months survival after AMI. It was found that during the 2008 study period, the 36 months survival among 25–54 and 55–64 year-old persons who experienced AMI was not significantly different, whereas in the other study periods, the survival rate of persons aged 25–54 was significantly better than those aged 55–64 years, 86.3% and 76.3% (*p* = 0.03), 89.4% and 82.6% (*p* = 0.009) and 92.3% and 79.8% (*p* = 0.003) in 1996, 2003–2004, and 2012 cohorts, respectively (Table 3).

The likelihood of long-term (36 months) survival in 25 to 64-year-old post-AMI Kaunas population in four time-periods (1996, 2003–2004, 2008, and 2012) is presented in Figure 1. According to the data of the Kaplan–Meier survival analysis, long-term survival of 25 to 64-year-old post-AMI persons was without significantly difference in 1996, 2003–2004, 2008, and 2012 (Log-rank = 6.736, *p* = 0.081).

The long-term (36 months) survival of subjects occurs of myocardial infarction in 1996, 2003–2004, 2008, and 2012 by age is shown in Figure 2 and Figure 3. Estimated, that long-term survival of 25 to 54-year-old post-AMI persons did not change significantly during the study period (Log-rank = 2.327, *p* = 0.507). The analysis of pairwise comparisons did not detect any significant changes in 36-month survival after AMI among 25 to 54-year-old Kaunas population in four time-periods. According to the data of the Kaplan-Meier survival analysis, 36 months survival of post-AMI persons aged 55–64 years was without significantly difference between 1996, 2003–2004, 2008, and 2012 time-periods too (Log-rank = 7.678, *p* = 0.053).

Hazard ratio of all-cause mortality in post-AMI patients during 36 months in Kaunas (Lithuania) in 1996, 2003–2004, 2008, and 2012 by sex and age are presented in Table 4. The crude risk of death from all causes in the sex groups did not show significant differences in comparison with the 1996 cohort as a reference with other cohorts of study. Evaluated the all-mortality risk in age groups, significant differences in the risk of death were identified only in the old age group (55–64 years). In 2008, the 55–64 age-old AMI persons had an average of 44% (HR = 0.56, 95% CI 0.36–0.88), a lower risk of death during the 36-month period compared to those who had experienced MI in 1996. The analysis of age-adjusted data showed analogous results.

In the Cox regression model, after adjusting data by sex, ICD-10 diagnosis and EDC significant changes in the risk of all-cause death within 36 months among post-AMI patients in 2003–2004, 2008, and 2012 compared to 1996, in both men and women, and in the age groups, were not found.

Evaluating the probability of all-cause death within 36 months among post-AMI in men and women, who had experienced AMI, adjusting by age, study cohort, ICD diagnosis, epidemiological diagnostic category, STEMI, previous acute heart failure, previous AMI, AH, stroke, diabetes, overweight and smoking, in 2003–2004, 2008, and 2012 cohorts compared to 1996 cohort, significant changes were not found. Assessing the risk of all-cause mortality within 36 months among post-AMI patients, who had experienced AMI, in the age groups adjusting data by sex, study cohort, ICD diagnosis, epidemiological diagnostic category, STEMI, previous acute heart failure, previous AMI, AH, stroke, diabetes, overweight, and smoking in 2003–2004, 2008, and 2012 cohorts compared to 1996 cohort significant changes were not found.

Factors predicting death from all-cause during 36 months in patients after AMI by sex are presented in Table 5. Multivariate survival analyses showed that among men older age (HR = 1.028, 95% CI 1.003–1.055), previous acute heart failure (HR = 6.255, 95% CI 4.335–9.024), previous AMI (HR = 1.503, 95% CI 1.070–2.113), stroke (HR = 1.877, 95% CI 1.030–3.422), diabetes (HR = 1.926, 95% CI 1.296–2.864) and obesity (HR = 1.884, 95% CI 1.070–3.318) were associated with worse 36 months survival after AMI, but overweight (HR = 0.671, 95% CI 0.473–0.953) with better survival during 36 months period after AMI. Among investigated women some confounders as STEMI (HR = 2.660, 95% CI 1.086–6.513), previous acute heart failure (HR = 9.166, 95% CI 4.217–19.923), stroke (HR = 4.964, 95% CI 1.843–13.374) and diabetes (HR = 3.873, 95% CI 1.961–7.648) were related with poorer 36 months survival after AMI, but overweight were associated with better 36 months survival after AMI (HR = 0.464, 95% CI 0.235–0.913).

Factors predicting fatal outcomes after AMI within a 36-month period from the onset of the disease depends on age are presented in Table 6. A multivariate survival analyses showed that among 25–54 years old patients only previous acute heart failure and stroke risk of death during a 36-month period after AMI increased by an average of 12 and 4 times, HR = 12.531, 95% CI 6.322–24.835 and HR = 3.966, 95% CI 1.316–11.953 respectively. Among 55–64 aged patients, some factors such as STEMI (HR = 1.807, 95% CI 1.152–2.834), previous acute heart failure (HR = 5.680, 95% CI 3.903–8.264), previous AMI (HR = 1.495, 95% CI 1.062–2.104), stroke (HR = 1.938, 95% CI 1.104–3.401), and diabetes (HR = 2.371, 95% CI 1.656–3.397) were related with worse 36 months survival after AMI, meanwhile overweight were associated with better 36 months survival after AMI, respectively HR = 0.623, 95% CI 0.431–0.901.

## 4. Discussion

In this study covering the period from 1996 to 2015, in which we analyzed long-term (36 months) survival among middle-aged patients who had experienced AMI in four time-periods—1996, 2003–2004, 2008, and 2012 noted tendency an increase in long-term survival of post-AMI patients during past years. Comparing the long-term survival rates after AMI, between 25–64 years old men and women in Kaunas did not receive significant differences in comparison of four study periods. The long-term survival rate among people who had experienced AMI improved during past years. Long-term survival after AMI among the younger age (25–54 years) and older age (55–64 years) group persons in 1996, 2003–2004, 2008, and 2012 did not change significantly too.

The data of declining in mortality rates from other countries could be attributed to many factors, including improved control of IHD risk factors, more common availability of invasive and non-invasive treatment methods (especially percutaneous coronary interventions (PCI)), and more frequent use of pharmacological agents [3,4,16,19]. In addition, the increase in the efficacy and safety of coronary artery stents may have improved the outcomes after AMI [22].

First of all, these trends could have determined due to better management of IHD risk factors and treatment changes after the first coronary event [17]. Despite the long-term survival improving among persons with AMI during past years, the level of risk factors for ischemic heart disease is still a concern because of the relatively high prevalence of these risk factors. Based on our findings, both the prevalence of AH and overweight, and even in both age groups, significantly increased during the last study periods. Based on other cohort epidemiological studies conducted in Kaunas, in which the risk factors for chronic non-infectious diseases were investigated, a similar level of both AH and overweight was found in a similar age population. In 2006–2008, in Kaunas city the prevalence of AH was 74.0% and 67.7% respectively, among 45–72 years-old persons, notwithstanding that a significant increase in AH awareness among hypertensive persons’ residents of Kaunas city and among treated hypertensive patients was observed during the analyzed period [Tamosiunas A., personal communication]. The frequency of AH and overweight among Kaunas men with acute coronary syndrome admitted to Department of Cardiology of the Lithuanian University of Health Sciences Hospital in 2007–2011 was 82.3% and 77.7%, respectively [23]. It should be noted, that in our study both AH and overweight were recorded according to the data in hospital health records when patients’ final clinical diagnosis was indicated. The frequency of other IHD risk factors such as DM has increased significantly over the last year, especially among younger (25–54 years) subjects, which may have led to the fact that long-term survival rates among younger people remained significantly unchanged compared to first time-period. Patients with DM have high long-term risk for AMI and risk of cardiovascular mortality is approximately 1.7-fold higher in this group, compared with those without DM [24].

Interestingly to note, that the impact of lifestyle and harmful risk factors on survival after an AMI, as well as the healthy eating habits of these individuals, cannot be ruled out. In the study, during four time-periods the prevalence of smoking among patients, who had experienced AMI, have tendency to decrease only in younger (25–54) age group, therefore in older patients, frequency of smoking did not change. In the EUROASPIRE III study, six months after AMI, smoking cessation was obtained in 48% of smokers, and regular exercise was performed by 34% of patients [25]. The EUROACTION and Blitz-4 Registry studies showed that a multidisciplinary cardiovascular disease secondary prevention programme may improve lifestyle changes and adherence to drug treatments in patients with coronary artery disease [26,27].

Secondly, in recent studies the researchers explored higher prevalence of non-STEMI among the elderly population. In our investigation, the frequency of STEMI among investigated persons with AMI did not change significantly during the study years as the frequency of STEMI in age groups was without significant changes, although the prevalence of STEMI in oldest (55–64 year) age group was lower in the last study periods. This may be due to the fact that in the last study period there were significantly more cases of non-STEMI in the older patients, troponin in addition to previously classical enzymes was introduced as a biomarker, which may have captured a smaller-sized AMI. We know that in recent years, the utilization of troponins has become widely available and that it could introduce many non-STEMI patients with smaller AMI who would have been diagnosed as unstable angina in the past. This finding could be related and partially explain improving survival post AMI among older patients in other studies [28,29,30].

Thirdly, much effort has been made to improve the management, diagnostics, and treatment of AMI, used modern secondary prevention measures after AMI during the last decades. With the leadership of the Lithuanian Society of Cardiology the measures of quality improvement for AMI have different aspects of management changes, including pre-hospital period and treatment procedures during hospitalization, and in period of outpatient care. In some surveys, accessibility to facilities of cardiac interventions procedures has improved too [31]. The management and healthcare system-related factors, such as lack of continuity of care, inadequate patient education, poor physician–patient relationships, can negatively impact adherence to medication after myocardial infarction [32,33].

Cardiac reperfusion rates are used as AMI treatment performance measures. The findings from recent studies indicate that reperfusion rates in Lithuanian hospitals are similar with the relevant rates from countries of West and North Europe [34,35]. As the first-line therapy for STEMI patients, primary PCI is recommended, but thrombolysis also persists as one of the strategies for reperfusion [36]. In addition to the reperfusion therapy, a major role in estimating the prognosis after AMI play the recommended related pharmacological therapy and the discharge drugs. By Thim T et al., among patients with AMI who discontinued antiplatelet drugs within 3 months after the PCI procedure mortality rates were 5 times higher to compare with patients who used these drugs 3 months [37]. In high-risk patients hospitalized with AMI and subsequently discharged on aspirin, ß-blockers, statin therapy, or a combination of all three medications, mortality was highest among patients who discontinued use of all medications (HR = 3.8, 95% CI 1.9–7.7). However, discontinuation of any of these treatments increased the risk of mortality, with nearly 2-fold and 3-fold increases seen with discontinuation of aspirin and statin therapies, respectively [38].

By some investigators, lower prevalence of prescription drugs for secondary prevention can be partly explained by differences in the baseline characteristics of AMI patients. Patients in the secondary care hospitals were older, and studies have shown that elderly patients are less likely to receive medications according guidelines [5,39]. Another reliable explanation could be lower compliance to the guidelines in smaller non-academic hospitals less frequently staffed with experienced specialists [40]. Therefore, the rates of the recommended drug use in tertiary care hospitals in Lithuania are similar to the relevant rates from the UK, Sweden, and the US [41]. In addition, the compliance of patients in using the recommended medications plays an important role in prognosis of survival. By some researchers, incapability to keep the proposed algorithms in drug use leads to more frequent hospital readmissions and has a negative influence on mortality rates [42,43]. Similar problems related to compliance in using the recommended medications after AMI have been estimated in some Baltic countries [17,44]. A similar large variation in AMI treatment methods between different hospitals and links with long-term mortality decreasing was previously estimated during 1996–2007 in Sweden [45] and from 1995 to 2006 in US [46].

Our study cannot exclude the potential impact of a cohort. In the 1996 cohort study, there could be more potential socioeconomic challenges for individuals due to transitional peculiarities in Lithuania than cohorts of later years of research. The particularities of certain infrastructural and organizational methods in the field of healthcare, which may have contributed to the survival of persons undergoing AMI in this research cohort, cannot be ruled out.

The multivariable analysis on the predictors of all-cause mortality after AMI provides some messages. It is interesting to note that if we evaluate the mortality risk and adjust mortality rates by age, we receive the reduction in mortality in the older (55–64 years) patients during the last survey period, except 2012 cohort, although if we estimate of mortality adjusting by gender, age, clinical diagnosis, EDC classification, STEMI, previous acute cardiac failure, previous MI, AH, stroke, DM, overweight, and smoking did not result in a significant reduction in mortality in the oldest age group. Meanwhile, the elimination from multivariable analysis model of sex, AH and smoking, and adjust of data from the above-mentioned factors, we got a significant reduction in the mortality rate in the older age group. Thus, control of AH and smoking in the older persons is important in reducing mortality after an AMI. In assessing the predictors of mortality in both sexes and younger age groups, no significant changes in mortality were obtained during the four study periods, adjusting mortality data only for age and all other factors mentioned above. Some researchers also note that controlling AH and smoking are important factors in improving the long-term survival of people with AMI [47,48]. It should be noted that during the study period, the proportion of men who had experienced a first AMI have tendency to decrease, except 2012 period, meanwhile the proportion of women increased during study period. In the assessment of long-term survival among AMI patients in the sex and age groups, some investigated factors were identified as determining 36 months survival after AMI without adjustment of time-years periods. A multivariate survival analyses determined that in men older age, previous acute heart failure, previous AMI, stroke, diabetes, and obesity were associated with worse, but overweight with better survival during 36 months period after AMI. Among investigated women STEMI, previous acute heart failure, stroke, and diabetes were related with poorer, but overweight were associated with better 36 months survival after AMI. Analyzing survival data in age groups was assessed that in 25–54 years old patients male gender, previous acute heart failure, and stroke significantly increased risk of death within 36 months, meanwhile among 55–64 aged patients STEMI, previous acute heart failure, previous AMI, stroke, and diabetes were related with worse, but overweight were associated with better 36 months survival after AMI. According to other researchers, women who had experienced AMI are less likely to control their blood pressure, body weight, more less physical active in comparison with men, and therefore their long-term survival is likely to be worse [27,47]. According to the data in Sweden and Norway, long-term mortality among some aged 28-day AMI survivors, women did not significantly worse than men when differences in comorbidities were considered [49,50].

The strengths of this study include its comparatively large sample size, the population-based level of representation, and validated data. In addition, we were capable of accurately tracking cases of death using National death registry data when reviewing death certificates.

### Limitations of the Study

The present study has several limitations. The first limitation is that we cannot prove the obvious reasons for the observed decrease causality in the survival rates. Secondly, the present study described four patient samples from the studied years but did not evaluate information about the previous events (first or recurrent) of AMI for the study cases. Thirsty, in the present study, we did not collect information about treatment regimens and contraindications to certain treatments. Fourthly, we did not collect data about medication use and the application of other secondary prevention methods including smoking cessation, alcohol consumption level, nutrition habits etc. Fifthly, we did not have data about comorbidities (stroke, DM, cancers, or respiratory and renal diseases) of post-AMI patients. Sixth, we were unable to assess the survival of elderly (65 years and older) in myocardial infarction due to the specificity of the study methodology. Finally, silent AMI that did not lead to hospital admission or death could not be captured by the available data and thus would not have been included in the study. Thus, we were unable to account for the effect of these or any other unmeasured confounders that might have influenced the long-term outcomes.

## 5. Conclusions

This study extends and updates current information on prognosis after AMI—specifically, long-term survival—among middle-aged Kaunas (Lithuania) population. It was found that 36 months survival among women and younger (25–54 years) persons was significant better compared to men and older (55–64 years) age persons. During the four study periods, despite the significantly increasing prevalence of major risk factors for ischemic heart disease, the long-term survival among 25 to 54-year-old post-AMI Kaunas population was without significantly changes, although it had a tendency to decrease among persons aged 55–64 years. Predictors of all-cause mortality after AMI during 36 months in men were older age, previous acute heart failure, previous AMI, stroke, and diabetes, in women STEMI, previous acute heart failure, stroke, and diabetes, and meanwhile in 25–54-year-olds, patients’ male gender, previous acute heart failure, and stroke and in 55–64 aged patients STEMI, previous acute heart failure, previous AMI, stroke and diabetes. The results highlight the fact that youngest age group remain a high-risk group, and reinforce the importance of primary and secondary prevention for the improvement of long-term prognosis of AMI patients.

## Figures and Tables

**Figure 1 medicina-55-00357-f001:**
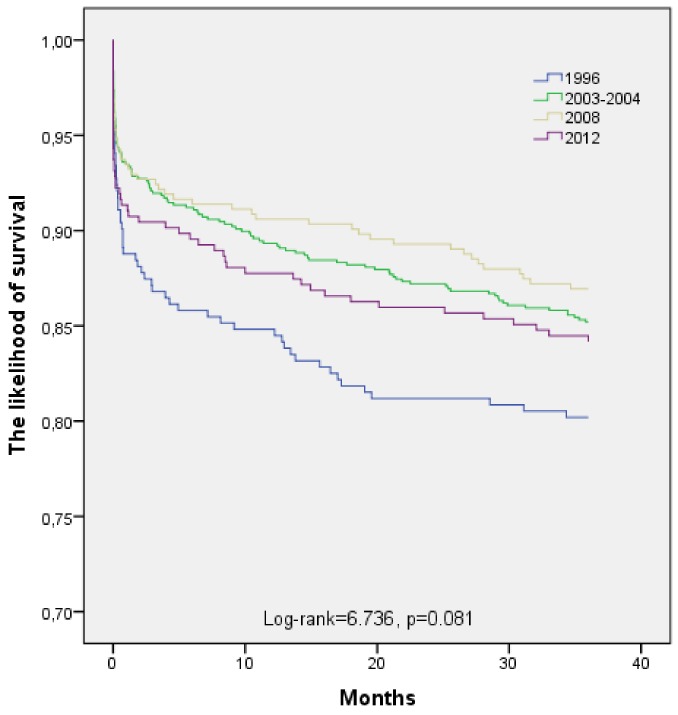
Long-term survival of patients aged 25–64 years with acute myocardial infarction by time-year’s periods.

**Figure 2 medicina-55-00357-f002:**
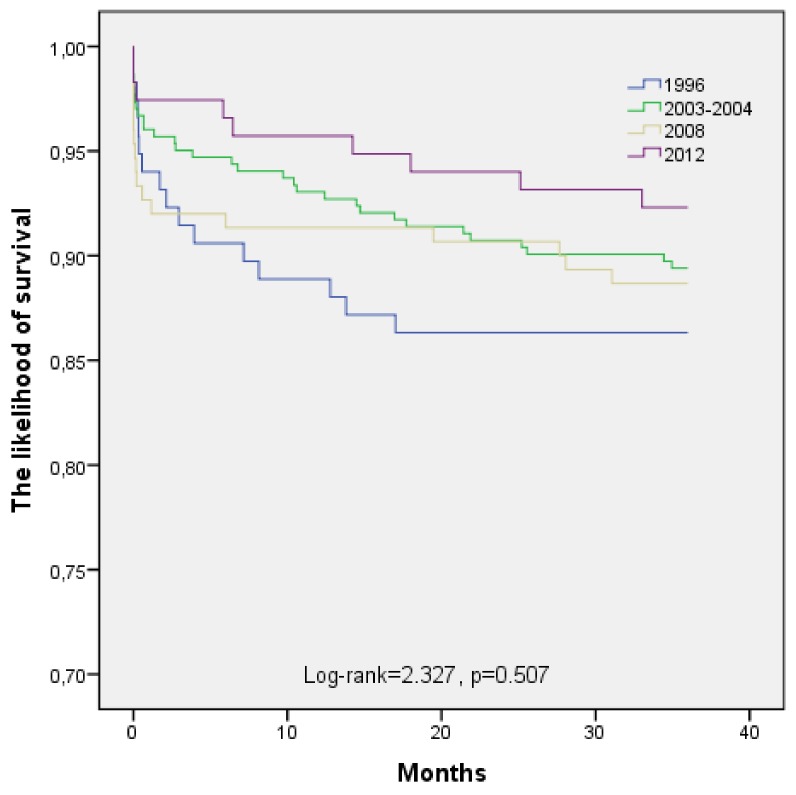
The long-term survival of patients aged 25–54 years with acute myocardial infarction by time-year’s periods.

**Figure 3 medicina-55-00357-f003:**
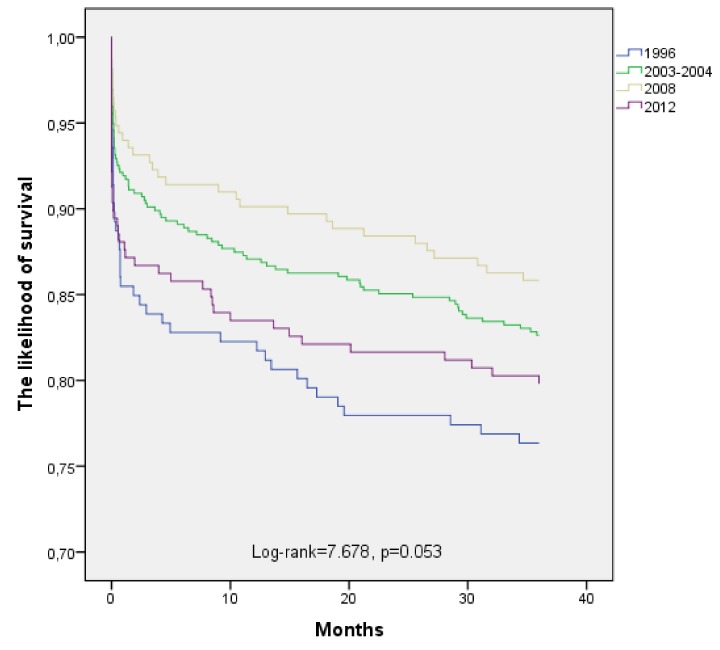
The long-term survival of patients aged 55–64 years with acute myocardial infarction by time-year’s periods.

**Table 1 medicina-55-00357-t001:** Baseline characteristics of patients by time-year periods.

Variables	1996	2003–2004	2008	2012	*p* for Trend
	N = 303	N = 797	N = 383	N = 335	
**Mean age (SD)**	55.3 (7.3)	55.0 (7.4)	55.6 (6.9)	56.0 (6.4)	0.26
**Men, %**	77.2 *	72.8	69.2	73.7	0.37
**55–64 years, %**	61.4	62.1	60.8	65.1	0.41
**Clinical diagnosis**					
**AMI**	75.6 *	60.0	54.6	61.2	0.23
**Unstable AP**	19.1 *	38.8	42.8	34.3	0.25
**EDC**					
**Definite AMI**	45.9 ^o^	47.7	51.7	55.8	0.04
**Possible AMI**	53.1 ^o^	51.7	47.0	39.4	0.1
**STEMI, %**	39.6	37.3	33.4	37.3	0.4
**AH, %**	65.7 *	71.0	80.2	83.0	0.02
**Stroke, %**	4.3	3.6	2.6	4.5	0.81
**Diabetes, %**	9.6 ^@^	14.4	13.8	17.0	0.07
**Previous AMI, %**	27.1	25.6	22.7	22.4	0.03
**Previous Acute heart failure, %**	10.2 *	6.8	6.0	6.3	0.1
**Smoking, %**	39.6 ^o^	37.3	38.9	30.1	0.26
**Overweight, %**	66.3	60.6 ^#^	67.6	63.9	0.95
**Hospital bed-days, mean, (SD)**	15.1 (9.4) ^&^	11.2 (7.0)	8.9 (5.5)	8.0 (6.1)	0.004

*—*p* < 0.05 compared 1996 year with 2008; ^#^—*p* < 0.05 compared 2003–2004 years with 2008; ^o^—*p* < 0.05 compared 1996 year with 2012; ^&^—*p* < 0.05 compared 1996 years with 2008 and 2012; ^@^—*p* < 0.05 compared 1996 year with 2003—2004 and 2012; ICD—international classification of diseases; EDC—epidemiological diagnostic category; AH—arterial hypertension; AMI—acute myocardial infarction; AP—angina pectoris; SD—standard deviation; STEMI—ST elevation myocardial infarction.

**Table 2 medicina-55-00357-t002:** Baseline characteristics of patients by age and time-year periods.

Variables/Age Group (Years)	1996	2003–2004	2008	2012	*p* for Trend
	N = 303	N = 797	N = 383	N = 335	
**Clinical diagnosis**					
**AMI**					
**25–54**	82.1 *	62.9	60.7	64.1	0.19
**55–64**	71.5 *	58.2	50.6	59.6	0.29
**Unstable AP**					
**25–54**	13.7 *	36.1	36.7	34.2	0.18
**55–64**	22.6 *	40.4	46.8	34.4	0.32
**EDC**					
**Definite AMI**					
**25–54**	43.6 *	49.7	54.7	57.3	0.004
**55–64**	47.3	46.5	49.8	55	0.17
**Possible AMI**					
**25–54**	56.4 *	50.3	43.3	41	0.009
**55–64**	51.1 ^@^	52.5	49.4	38.5	0.25
**STEMI, %**					
**25–54**	37.6	38.7	38.7	35.9	0.6
**55–64**	40.9 *	36.4	30	38.1	0.52
**AH, %**					
**25–54**	69.2 *	69.5	78	85.5	0.09
**55–64**	63.4 *	71.9	81.5	81.7	0.03
**Stroke, %**					
**25–54**	1.7	2.6#	0.7	2.6	0.96
**55–64**	5.9	4.2	3.9	5.5	0.7
**Diabetes, %**					
**25–54**	3.4 *^o^	9.3	10.2	13.9	0.049
**55–64**	13.7	17.8	16.5	20	0.12
**Previous AMI, %**					
**25–54**	21.4 *	16.2	16	14.5	0.05
**55–64**	30.6	31.3	27	26.6	0.16
**Previous acute heart failure, %**					
**25–54**	6.8	6.3	4	3.4	0.06
**55–64**	12.4 ^o^	7.1	7.3	7.8	0.24
**Smoking, %**					
**25–54**	56.4 *	52	48	40.2	0.05
**55–64**	29	28.3	33	24.8	0.72
**Overweight, %**					
**25–54**	62.4	60.3 ^#^	68.7	69.2	0.21
**55–64**	68.8 ^o^	60.8	67	61	0.44
**Hospital bed-days, mean, (SD)**					
**25–54**	14.1 (5.9) *	10.6 (6.6)	8.3 (4.9)	8.1 (6.4)	0.02
**55–64**	15.8 (10.9) *	11.3 (7.2)	9.2 (5.7)	8.0 (6.0)	0.002

*—*p* < 0.05 compared 1996 year with 2008 and 2012 years; ^#^—*p* < 0.05 compared 2003–2004 years with 2008 and 2012 years; ^o^—*p* < 0.05 compared 1996 year with 2003–2004 years; ^@^—*p* < 0.05 compared 1996 year with 2008; ICD—international classification of diseases; EDC—epidemiological diagnostic category; AH—arterial hypertension; AMI—acute myocardial infarction; AP—angina pectoris; SD—standard deviation, STEMI—ST elevation myocardial infarction.

**Table 3 medicina-55-00357-t003:** The distribution of Kaunas 25 to 64-year-old patients with acute myocardial infarction by sex, age, time-year periods, and survival time.

Sex and Age Groups	Study Year	Total N	N of Death	Survived		Means of Survival Time in Months		
						Estimate	95% CI	
				N	%		Lower	Upper
**Men**	**1996**	234	47	187	79.9	29.9	28.5	31.4
	**2003–2004**	580	91	489	84.3	31.9	31.2	32.6
	**2008**	265	39	226	85.3	32.4	31.4	33.4
	**2012**	247	43	204	82.6	30.9	29.4	32.4
	**Overall**	1326	220	1106	83.4	31.2	30.6	31.8
**Women**	**1996**	69	13	56	81.2	30.7	27.8	33.5
	**2003–2004**	217	27	190	87.6	32.6	31.3	33.9
	**2008**	118	11	107	90.7	33.6	32.1	35.2
	**2012**	88	10	78	88.6	32.5	30.4	34.7
	**Overall**	492	61	431	87.6	32.6	31.7	33.5
**25–54 Years**	**1996**	117	16	101	86.3	31.7	29.7	33.7
	**2003–2004**	302	32	270	89.4	33.2	32.2	34.2
	**2008**	150	17	133	88.7	32.7	31.1	34.3
	**2012**	117	9	108	92.3	34.1	32.8	35.4
	**Overall**	686	74	612	89.2	33.0	32.3	33.7
**55–64 Years**	**1996**	186	44	142	76.3	28.8	26.9	30.8
	**2003–2004**	495	86	409	82.6	31.1	30.2	32.1
	**2008**	233	33	200	85.8	32.2	30.9	33.5
	**2012**	218	44	174	79.8	29.9	28.1	31.6
	**Overall**	1132	207	925	81.7	30.7	30.0	31.4
**Total**	**1996**	303	60	243	80.2	29.9	28.5	31.4
	**2003–2004**	797	118	679	85.2	31.9	31.2	32.6
	**2008**	383	50	333	86.9	32.4	31.4	33.4
	**2012**	335	53	282	84.2	31.4	30.1	32.6
	**Overall**	1818	281	1537	84.5	31.6	31.1	32.1

95% CI—95% confidence interval.

**Table 4 medicina-55-00357-t004:** Hazard ratio of all-cause mortality in patients during 36 months after acute myocardial infarction by sex, age, and time-year period (Cox regression analysis).

Mortality	Sex		Age Group (Years)	
	Men	Women	25–54	55–64
**HR (95% CI)**				
**1996**	1	1	1	1
**2003–2004**	0.754 (0.53–1.072)	0.631 (0.325–1.222)	0.756 (0.415–1.377)	0.70 (0.487–1.007)
**2008**	0.707 (0.463–1.081)	0.464 (0.208–1.037)	0.824 (0.416–1.63)	0.559 (0.356–0.878)
**2012**	0.855 (0.565–1.293)	0.584 (0.256–1.332)	0.540 (0.239–1.222)	0.845 (0.556–1.283)
**p trend**	0.51	0.16	0.21	0.4
**Adjusted HR (95% CI) ***				
**1996**	1	1	1	1
**2003–2004**	0.762 (0.536–1.083)	0.652 (0.336–1.264)	0.765 (0.42–1.395)	0.708 (0.493–1.019)
**2008**	0.692 (0.453–1.058)	0.486 (0.218–1.085)	0.812 (0.41–1.607)	0.554 (0.353–0.870)
**2012**	0.833 (0.551–1.26)	0.606 (0.266–1.384)	0.520 (0.23–1.179)	0.857 (0.564–1.302)
**p trend**	0.4	0.18	0.17	0.4
**Adjusted HR (95% CI) ****				
**1996**	1	1	1	1
**2003–2004**	1.067 (0.731–1.556)	0.922 (0.468–1.817)	1.220 (0.623–2.392)	0.961 (0.657–1.406)
**2008**	0.968 (0.621–1.507)	0.606 (0.266–1.384)	0.988 (0.463–2.109)	0.798 (0.501–1.270)
**2012**	0.868 (0.550–1.371)	0.732 (0.314–1.709)	0.647 (0.261–1.601)	0.891 (0.570–1.393)
**p trend**	0.41	0.28	0.31	0.44
**Adjusted HR (95% CI) *****				
**1996**	1	1		
**2003–2004**	1.329 (0.858–2.058)	1.193 (0.513–2.775)		
**2008**	1.241 (0.737–2.089)	1.211 (0.388–3.784)		
**2012**	1.219 (0.690–2.153)	1.088 (0.342–3.461)		
**p trend**	0.62	0.90		
**Adjusted HR (95% CI) ******				
**1996**			1	1
**2003–2004**			1.176 (0.536–2.578)	1.403 (0.904–2.178)
**2008**			1.796 (0.735–4.391)	1.203 (0.693–2.089)
**2012**			0.894 (0.312–2.561)	1.350 (0.760–2.399)
**p trend**			0.82	0.47

HR—hazard ratio, 95% CI—95% confidence interval * Adjusted by age ** Adjusted by age, study cohort, International Classification of Diseases diagnosis, epidemiological diagnostic category *** Adjusted by age, study cohort, International Classification of Diseases diagnosis, epidemiological diagnostic category, ST elevated myocardial infarction, previous acute heart failure, previous acute myocardial infarction (AMI), arterial hypertension (AH), stroke, diabetes, overweight and smoking **** Adjusted by sex, study cohort, ICD diagnosis, epidemiological diagnostic category, ST elevated myocardial infarction, previous acute heart failure, previous AMI, AH, stroke, diabetes, overweight and smoking.

**Table 5 medicina-55-00357-t005:** Hazard ratio of all-cause mortality during 36 months in patients after acute myocardial infarction by sex (Cox regression analysis).

Variables	B	HR	95% CI		*p*
Men					
Time-year period (every 1 period)	0.043	1.044	0.881	1.237	0.619
Age (every 1 year)	0.028	1.028	1.003	1.055	0.029
AMI (ref.)	1				
Unstable AP	−0.391	0.676	0.394	1.162	0.156
Definite AMI (ref.)	1				
Possible AMI	−0.186	0.830	0.528	1.306	0.420
STEMI (ref. no)	0.222	1.249	0.827	1.886	0.292
Previous acute heart failure (ref. no)	1.833	6.255	4.335	9.024	0.0001
Previous AMI (ref. no)	0.408	1.503	1.070	2.113	0.019
AH (ref. no)	−0.176	0.838	0.589	1.194	0.329
Stroke (ref. no)	0.630	1.877	1.030	3.422	0.040
Diabetes (ref. no)	0.656	1.926	1.296	2.864	0.001
Normal weight (ref.)	1				
Overweight	−0.399	0.671	0.473	0.953	0.026
Obesity	0.634	1.884	1.070	3.318	0.028
Smoking (ref. never smokers)	0.098	1.103	0.794	1.532	0.559
**Women**					
Time-year period (every 1 period)	0.024	1.024	0.714	1.469	0.896
Age (every 1 year)	0.056	1.057	0.990	1.129	0.095
AMI (ref.)	1				
Unstable AP	−0.082	0.921	0.329	2.577	0.876
Definite AMI (ref.)	1				
Possible AMI	0.066	1.068	0.478	2.388	0.873
STEMI (ref. no)	0.978	2.660	1.086	6.513	0.032
Previous acute heart failure (ref. no)	2.215	9.166	4.217	19.923	0.0001
Previous AMI (ref. no)	−0.044	0.957	0.455	2.011	0.908
AH (ref. no)	−0.548	0.578	0.281	1.188	0.136
Stroke (ref. no)	1.602	4.964	1.843	13.374	0.002
Diabetes (ref. no)	1.354	3.873	1.961	7.648	0.0001
Normal weight (ref.)	1				
Overweight	−0.769	0.464	0.235	0.913	0.026
Obesity	−0.538	0.584	0.201	1.699	0.324
Smoking (ref. no)	−1.357	0.257	0.034	1.950	0.189

HR—hazard ratio; B—B coefficient; 95% CI—95% confidence interval; *p*—significance level STEMI—ST elevated myocardial infarction; AMI—acute myocardial infarction; AP—angina pectoris; AH—arterial hypertension; ref.—reference (comparative) group; ref. no—reference (comparative) group without the indicated disorders.

**Table 6 medicina-55-00357-t006:** Hazard ratio of all-cause mortality during 36 months in patients after acute myocardial infarction by age (Cox regression analysis).

	B	HR	95% CI		*p*
**25–54 age group**					
Time-year period (every 1 period)	0.008	1.008	0.753	1.350	0.956
Age (every 1 year)	0.052	1.053	0.994	1.115	0.077
Sex (ref. men)	−1.014	0.363	0.132	0.996	0.049
AMI (ref.)	1				
Unstable AP	−0.087	0.916	0.350	2.396	0.859
Definite AMI (ref.)	1				
Possible AMI	−0.491	0.612	0.265	1.409	0.248
STEMI (ref. no)	−0.186	0.830	0.408	1.690	0.608
Previous acute heart failure (ref. no)	2.528	12.531	6.322	24.835	0.0001
Previous AMI (ref. no)	0.182	1.20	0.596	2.415	0.610
AH (ref. no)	−0.133	0.875	0.480	1.597	0.664
Stroke (ref. no)	1.378	3.966	1.316	11.953	0.014
Diabetes (ref. no)	0.532	1.702	0.692	4.183	0.247
Normal weight (ref.)	1				
Overweight	−0.606	0.546	0.298	1.000	0.050
Obesity	0.580	1.785	0.655	4.869	0.257
Smoking (ref. no)	0.084	1.088	0.582	2.035	0.792
**55–64 age group**					
Time-year period (every 1 period)	0.064	1.066	0.894	1.271	0.475
Age (every 1 year)	0.014	1.014	0.958	1.074	0.633
Sex (ref. men)	−0.119	0.888	0.595	1.325	0.561
AMI (ref.)	1				
Unstable AP	−0.422	0.656	0.376	1.144	0.137
Definite AMI (ref.)	1				
Possible AMI	−0.026	0.975	0.619	1.535	0.912
STEMI (ref. no)	0.592	1.807	1.152	2.834	0.010
Previous acute heart failure (ref. no)	1.737	5.680	3.903	8.264	0.0001
Previous AMI (ref. no)	0.402	1.495	1.062	2.104	0.021
AH (ref. no)	−0.328	0.721	0.497	1.046	0.084
Stroke (ref. no)	0.661	1.938	1.104	3.401	0.021
Diabetes (ref. no)	0.863	2.371	1.656	3.397	0.0001
Normal weight (ref.)	1				
Overweight	−0.473	0.623	0.431	0.901	0.012
Obesity	0.234	1.264	0.708	2.256	0.429
Smoking (ref. no)	−0.055	0.946	0.656	1.365	0.768

HR—hazard ratio; B—B coefficient; 95% CI—95% confidence interval; *p*—significance level STEMI—ST elevated myocardial infarction; AMI—acute myocardial infarction; AP—angina pectoris; AH—arterial hypertension; ref.—reference (comparative) group; ref. no—reference (comparative) group without the indicated disorders.

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
