# Peer review of "Long-Term Survival after Acute Myocardial Infarction in Lithuania during Transitional Period (1996–2015): Data from Population-Based Kaunas Ischemic Heart Disease Register"

_medicina, 2019, doi:10.3390/medicina55070357_

Round 1
Reviewer 1 Report
All concerned issues have responsed by authors. I have no other comment.
Reviewer 2 Report
I don't have comments and suggestions for authors.
This manuscript is a resubmission of an earlier submission. The following is a list of the peer review reports and author responses from that submission.
Round 1
Reviewer 1 Report
In this study, the authors investigated the rates of long-term survival after acute myocardial infarction, in Lithuania (Kaunas) population in four times periods 1996, 2003-2004, 2008 and 2012. The results show that the long-term survival rate among people post-AMI improved during past years.
However, there are some issues concerning the analysis and presentation of data that the authors need to correct and clarify in order to give a clear and informative message to the readers.
How do You explain the statement: risk factors for ischemic heart disease increased in 2012, although survival after-AMI was better in past period?
2. The conclusions should better represent the results.
3. Mistakes in the table 2 should be corrected.
Author Response
Dear Reviewer
We would like to thank you for your valuable comments regarding our manuscript “Long-term survival after acute myocardial infarction in Lithuania during transitional period (1996-2015): data from population-based Kaunas ischemic heart disease register” submitted to Medicina journal (Manuscript ID: medicina-419125).
We hope that the quality of our manuscript was improved as suggested. All modifications in the text of the manuscript and tables are in „yellow“.
At the bottom, we attach our responses to the received reviewer comments.
Reviewer 1
In this study, the authors investigated the rates of long-term survival after acute myocardial infarction, in Lithuania (Kaunas) population in four times periods 1996, 2003-2004, 2008 and 2012. The results show that the long-term survival rate among people post-AMI improved during past years.
However, there are some issues concerning the analysis and presentation of data that the authors need to correct and clarify in order to give a clear and informative message to the readers.
How do You explain the statement: risk factors for ischemic heart disease increased in 2012, although survival after-AMI was better in past period?
Response: Statement, that risk factors for ischemic heart disease increased in 2012, although survival after-AMI was better in past period we could explain it by better outpatient care for persons with myocardial infarction, better diagnostic and treatment algorithms in recent years, more responsible patient behavior while taking medication and rehabilitation procedures, meanwhile that for the last study period (2012-2015) 36 months survival after MI was worse.
2. The conclusions should better represent the results.
Response: We corrected and strengthened our conclusions by providing more specific data.
3. Mistakes in the table 2 should be corrected.
Response: Mistakes in Table 2 were corrected.
Reviewer 2 Report
The authors aim to evaluate the long-term (36 months) survival after AMI among persons aged 25–64 years, who survived at least 28 days and who had experienced AMI in four time-periods 1996, 2003–2004, 2008 17 and 2012 based on Kaunas population-based Ischemic heart 18 disease (IHD) register. The authors found long-term survival among 25-64 year-old post-AMI Kaunas population was without significantly changes and concluded that AMI survivors remain a high-risk group and reinforce the importance of primary and secondary prevention for the improvement of long-term prognosis of AMI patients.
This topic is interesting because report regarding the epidemiological study on the long-term survival after AMI in Lithuania population is limited. I believed this study might informative for the health care specialists in Lithuania.
However, I do have several comments that I believe warrant further addressing, especially about the methodology of the current study:
Major comments:
1. Authors defined 36 months as the long-term survival after AMI, please explain the reason for using this time point as the cutoff value in your study and cite reference to support your claim
2. Why authors only focus on 25-64 year-old population in this study. Please address the reason for age selection in the introduction section. Why authors categorize patients into 25-54 and 55-64 years for further comparison in the table and Figure 2 and Figure 3.
Minor comments:
1. I am confused the meaning of p value in all of the tables. For example, in Table 1, men were 77.2% in 1996 and were 68.2% in 2008. The authors using “*” following value of 77.2% and 68.2% to present a significant statistical difference compared 1996 year with other years. If authors using 1996 as the reference year, what does the meaning of “77.2%*” indicating as it compare to itself ?
2. some numbers in Table 2 is missing, for example, DM, %, 5-54; previous MI, % 5-64.
3. Table 3, means of survival time is confused to me because more than half of the patients still alive at end of study, I believe the mean survival time must longer than 36 months.
4. A rapid drop survival time immediately (almost 1-2 months at the beginning of the survival curve) after AMI across 4 study time period was noted in Figure 1 and Figure 3. I belief authors had excluded patients died within 28 days in this study. Please explain the result because too number patients died in the acute phase inevitable impact the long-term outcome of these patients group.
Author Response
Dear Reviewer
We would like to thank you for your valuable comments regarding our manuscript “Long-term survival after acute myocardial infarction in Lithuania during transitional period (1996-2015): data from population-based Kaunas ischemic heart disease register” submitted to Medicina journal (Manuscript ID: medicina-419125).
We hope that the quality of our manuscript was improved as suggested. All modifications in the text of the manuscript and tables are in „yellow“.
At the bottom, we attach our responses to the received reviewer comments.
Reviewer 2
The authors aim to evaluate the long-term (36 months) survival after AMI among persons aged 25–64 years, who survived at least 28 days and who had experienced AMI in four time-periods 1996, 2003–2004, 2008 17 and 2012 based on Kaunas population-based Ischemic heart 18 disease (IHD) register. The authors found long-term survival among 25-64 year-old post-AMI Kaunas population was without significantly changes and concluded that AMI survivors remain a high-risk group and reinforce the importance of primary and secondary prevention for the improvement of long-term prognosis of AMI patients.
This topic is interesting because report regarding the epidemiological study on the long-term survival after AMI in Lithuania population is limited. I believed this study might informative for the health care specialists in Lithuania.
However, I do have several comments that I believe warrant further addressing, especially about the methodology of the current study:
Major comments:
1. Authors defined 36 months as the long-term survival after AMI, please explain the reason for using this time point as the cutoff value in your study and cite reference to support your claim
Response: The authors of the manuscript selected a 36-month survival period as a long-term survival after MI to compare survival rates in 4 periods in Lithuania during the transition period, which is a sufficiently long study period to monitor deaths events. A 36-month period after myocardial infarction is a long enough follow-up period to explain some of the nuances associated with controlling myocardial infarction by some sociodemographical and risk factors, management, diagnosis, treatment and rehabilitation peculiarities. A longer follow-up period (e.g. 5, 7, or 10 years) would be complicated for registration of all death events due to greater data loss over a longer period of follow-up, possibility for more incomplete data analysis, possibilities of some other comorbidities as causes of death due to longer follow-up period as well as changes in diagnostics, treatment and rehabilitation algorithms. Some researchers also provide tracking data for persons who have had myocardial infarction using a 36 months follow-up period as sufficiently long time period to evaluate survival rates after myocardial infarction [references added].
Reference:
1. Puymirat E, Taldir G, Aissaoui N, Lemesle G, Lorgis L, Cuisset T et al. Use of invasive strategy in non-ST-segment elevation myocardial infarction is a major determinant of improved long-term survival: FAST-MI (French Registry of Acute Coronary Syndrome). JACC Cardiovasc Interv 2012;5(9):893-902. doi: 10.1016/j.jcin.2012.05.008.
2. Puymirat E, Aissaoui N, Silvain J, Bonello L, Cuisset T, Motreff P et al.; FAST-MI investigators. Comparison of bleeding complications and 3-year survival with low-molecular-weight heparin versus unfractionated heparin for acute myocardial infarction: the FAST-MI registry. Arch Cardiovasc Dis 2012;105(6-7):347-54. doi: 10.1016/j.acvd.2012.04.002. Epub 2012 Jun 27.
3. Alapati V, Tang F, Charlap E, Chan PS, Heidenreich PA, Jones PG et al. Discharge Heart Rate After Hospitalization for Myocardial Infarction and Long-Term Mortality in 2 US Registries. JAMA 2014;312(23):2510-20. doi: 10.1001/jama.2014.15690.
4. Hou LL, Gao C, Feng J, Chen ZF, Zhang J, Jiang YJ et al. Prognostic Factors for In-Hospital and Long-Term Survival in Patients with Acute ST-Segment Elevation Myocardial Infarction after Percutaneous Coronary Intervention. Tohoku J Exp Med 2017;242(1):27-35. doi: 10.1620/tjem.242.27.
5. Roos A, Sartipy U, Ljung R, Holzmann MJ. Relation of Chronic Myocardial Injury and Non-ST-Segment Elevation Myocardial Infarction to Mortality. Am J Cardiol 2018;122(12):1989-1995. doi: 10.1016/j.amjcard.2018.09.006. Epub 2018 Sep 15.
6. Acharya T, Aspelund T, Jonasson TF, Schelbert EB, Cao JJ, Sathya B et al. Association of Unrecognized Myocardial Infarction With Long-term Outcomes in Community-Dwelling Older Adults: The ICELAND MI Study. JAMA Cardiol 2018;3(11):1101-1106. doi: 10.1001/jamacardio.2018.3285.
7. Fu WX, Zhou TN, Wang XZ, Zhang L, Jing QM, Han YL. Sex-Related Differences in Short- and Long-Term Outcome among Young and Middle-Aged Patients for ST-Segment Elevation Myocardial Infarction Underwent Percutaneous Coronary Intervention. Chin Med J (Engl) 2018;131(12):1420-1429. doi: 10.4103/0366-6999.233965.
2. Why authors only focus on 25-64 year-old population in this study. Please address the reason for age selection in the introduction section. Why authors categorize patients into 25-54 and 55-64 years for further comparison in the table and Figure 2 and Figure 3.
Response: According to the MONICA project recommendations and Ischemic heart disease register, until recent years’ only persons of working age (25-64 years) were included in the Ischemic heart disease register, therefore it was not possible to assess the long-term survival rates of elderly persons (65 and over years). This was seen as one of the limitations of our study. While emphasizing that long-term survival of myocardial infarction was investigated in working-age (25-64 years) persons, we added this point in the introduction section. The manuscript authors categorize patients into 25-54 and 55-64 years age groups for further comparison, because there were very small numbers of events in youngest age groups, especially in 25-34 years and 35-44 years age groups, so we had to combine the younger groups into one larger youngest group (25-54 years) and one older group (55-64 years) to have approximately the same group of subjects. Combined age groups (35-54 and 55-64) were also investigated by other researchers [reference added].
Reference:
Berg J, Björck L, Nielsen S, Lappas G, Rosengren A. Sex differences in survival after myocardial infarction in Sweden, 1987–2010. Heart 2017;103:1625–1630.
Minor comments:
1. I am confused the meaning of p value in all of the tables. For example, in Table 1, men were 77.2% in 1996 and were 68.2% in 2008. The authors using “*” following value of 77.2% and 68.2% to present a significant statistical difference compared 1996 year with other years. If authors using 1996 as the reference year, what does the meaning of “77.2%*” indicating as it compare to itself ?
Response: Meanings of p values in all tables were corrected.
2. some numbers in Table 2 is missing, for example, DM, %, 5-54; previous MI, % 5-64.
Response: Mistakes in Table 2 in DM and previous MI sectors were corrected.
3. Table 3, means of survival time is confused to me because more than half of the patients still alive at end of study, I believe the mean survival time must longer than 36 months.
Response: This study assessed 36 months long-term survival after myocardial infarction, allowing people who have lived for 36 months to survive hypothetically higher. According survival analysis evaluation methods, an average survival in 36 months after MI may could not be longer than 36 months, because end cut-off point for survival tracking was 36 months.
4. A rapid drop survival time immediately (almost 1-2 months at the beginning of the survival curve) after AMI across 4 study time period was noted in Figure 1 and Figure 3. I belief authors had excluded patients died within 28 days in this study. Please explain the result because too number patients died in the acute phase inevitable impact the long-term outcome of these patients group.
Response: In the Figures, we presented Kaplan-Meier Survival Analysis data showing survival data from the onset of myocardial infarction, but the survival data were calculated and evaluated only among persons who survived first 28 days from the onset of myocardial infarction.

Reviewer 3 Report
Thank you for the possibility to review this manuscript.
In this manuscript authors describe the long term survival after AMI in lithuanian population. I find the study appropriate and with interest for the readers.
I have only some minor questions:
a) why did authors exluded elderly population? What about the patietns of more than 65 years old?
b) how many patients were lost or with uncomplete data?
c) the authors describe in baseline chactersitics: previous myocardal infartion, hypertension,stroke, diabetes and smoking. Are there any data about dislipidemia?
Author Response
Dear Reviewer
We would like to thank you for your valuable comments regarding our manuscript “Long-term survival after acute myocardial infarction in Lithuania during transitional period (1996-2015): data from population-based Kaunas ischemic heart disease register” submitted to Medicina journal (Manuscript ID: medicina-419125).
We hope that the quality of our manuscript was improved as suggested. All modifications in the text of the manuscript and tables are in „yellow“.
At the bottom, we attach our responses to the received reviewer comments.
Reviewer 3
Thank you for the possibility to review this manuscript.
In this manuscript authors describe the long term survival after AMI in lithuanian population. I find the study appropriate and with interest for the readers.
I have only some minor questions:
1. why did authors exluded elderly population? What about the patietns of more than 65 years old?
Response: According to the MONICA project recommendations and Ischemic heart disease register, until recent years’ only persons of working age (25-64 years) were included in the Ischemic heart disease register, therefore it was not possible to assess the long-term survival rates of elderly persons (65 and over years). This was seen as one of the limitations of our study, which we pointed out in the section on the limitation of the study.
2. how many patients were lost or with uncomplete data?
Response: According to our survey, during 4 time periods at the end of follow-up 42 (2.3%) persons were lost to follow-up and estimated 26 (1.4%) cases with incomplete data.
3. the authors describe in baseline chactersitics: previous myocardal infartion, hypertension,stroke, diabetes and smoking. Are there any data about dislipidemia?
Response: Currently, the Ischemic heart disease register has started collecting data on dyslipidemias in persons with MI, but unfortunately these data has not been collected since the beginning of our study. This could be one of the limitation of our investigation.

Round 2
Reviewer 2 Report
Thanks authors' effort to improve the performance of this manuscript. I have no further comments.
Author Response
Dear Reviewer,
We would like to thank you for your valuable comments regarding our manuscript “Long-term survival after acute myocardial infarction in Lithuania during transitional period (1996-2015): data from population-based Kaunas ischemic heart disease register” submitted to Medicina journal (Manuscript ID: medicina-419125).